# Economics of healthcare access in low-income and middle-income countries: a protocol for a scoping review of the economic impacts of seeking healthcare on slum-dwellers compared with other city residents

Noemia Teixeira de Siqueira-Filha [1], Jinshuo Li [1], Eliud Kibuchi [2], Zahidul Quayyum,[3] Penelope Phillips-Howard,[4] Abdul Awal,[3] Md Imran Hossain Mithu [3], Farzana Manzoor,[3] Robinson Karuga,[5] Samuel Saidu,[6] John Smith,[6] Varun Sai,[7] Sureka Garimella,[7] Ivy Chumo,[8] Blessing Mberu,[8] Rachel Tolhurst,[9] Sumit Mazumdar [10], Vinodkumar Rao,[11] Nadia Farnaz,[12] Wafa Alam,[3] Helen Elsey[1]

► Prepublication history and additional online supplemental material for this paper are available online. To view these files, please visit the journal online (http://dx.doi.org/10.1136/bmjopen-2020-045441).

For numbered affiliations see end of article.

**Correspondence to**
Dr Noemia Teixeira de Siqueira-Filha;
noemia.teixeira.siqueira@gmail.com

## ABSTRACT

**Introduction** People living in slums face several challenges to access healthcare. Scarce and low-quality public health facilities are common problems in these communities. Costs and prevalence of catastrophic health expenditures (CHE) have also been reported as high in studies conducted in slums in developing countries and those suffering from chronic conditions and the poorest households seem to be more vulnerable to financial hardship. The COVID-19 pandemic may be aggravating the economic impact on the extremely vulnerable population living in slums due to the long-term consequences of the disease. The objective of this review is to report the economic impact of seeking healthcare on slum-dwellers in terms of costs and CHE. We will compare the economic impact on slum-dwellers with other city residents.

**Methods and analysis** This scoping review adopts the framework suggested by Arksey and O'Malley. The review is part of the accountability and responsiveness of slum-dwellers (ARISE) research consortium, which aims to enhance accountability to improve the health and well-being of marginalised populations living in slums in India, Bangladesh, Sierra Leone and Kenya. Costs of accessing healthcare will be updated to 2020 prices using the inflation rates reported by the International Monetary Fund. Costs will be presented in International Dollars by using purchase power parity. The prevalence of CHE will also be reported.

**Ethics and dissemination** Ethical approval is not required for scoping reviews. We will disseminate our results alongside the events organised by the ARISE consortium and international conferences. The final manuscript will be submitted to an open-access international journal. Registration number at the Research Registry: reviewregistry947.

### Strengths and limitations of this study

► This is a comprehensive scoping review that includes studies addressing costs of illness and cost of accessing healthcare to identify inequities within the urban context.
► There is no language restriction in this review, peer-reviewed studies, grey literature and reports from 12 databases including leading organisations on urban health research will be searched.
► We will assess the quality of studies by using the Consolidated Health Economic Evaluation Reporting Standards and the Tool to Estimate Patient's Costs.
► Slums communities in rural areas will not be included in this review.

## BACKGROUND

The United Nations estimates that an increase in population and migration will add a further 2.5 billion people to the urban population by 2050. Ninety per cent of this growth will occur in Asian and African countries.[1] In 2014, 30% of the total urban population in developing countries were living in slums.[2] The absolute number of 881 080 000 people is close to the entire population of the two most populous countries in the world, India (1.4 billion inhabitants) and China (1.3 billion inhabitants).[3] While urbanisation is one possible driver of economic growth,[4] it also increases inequities within cities and those living in slum households and neighbourhoods suffer grossly inadequate access to

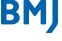

healthcare, clean water, sanitation, safe housing and basic amenities.[5]

Although the traditional assumption describes an urban advantage in accessing healthcare,[6] many barriers to access remain, and physical proximity to health providers does not always translate to better health, particularly for the urban poor.[7] While analysis of Demographic and Health Surveys (DHSs) across 73 low-income and middle-income countries (LMICs) showed that children in urban slums have better health than their rural counterparts,[8] this masks intraurban inequities. For example, under 5 mortality in Kenya and Bangladesh was higher in urban slums than in rural areas (Kenya: 79.8 vs 56 per 1000 live births; Bangladesh: 57 vs 49 per 1000 live births).[7 9] These conflicting data may in part be due to limitations in traditional sampling methods used by DHS and other national surveys where urban poor are frequently under-represented.[10 11]

Urbanisation has been seen as a determinant of health and well-being, resulting in a double burden of communicable and non-communicable diseases (NCDs).[12] A spectrum of research has shown how slums constitute unhealthy places, posing critical health challenges for urban residents. Children living in slums are particularly vulnerable to malnutrition, recurrent diarrhoea, stunted growth and long-term cognitive development challenges.[13 14] Prevalence of diabetes and hypertension,[6 15] and poor mental health have increased dramatically in slums.[7 8] Injuries, both unintentional and as a result of violence,[16] nutrition transition,[17 18] obesity,[19] tobacco[20 21] and alcohol abuse[22 23] are also disproportionately high in slum communities. All of these studies highlight how these poor health conditions are driven by the stresses, work patterns and changing social and gender norms within cities. There is also increasing evidence on the role of socioeconomic factors contributing to common forms of mental illness in slum-dwellers, further exacerbating the burden of disease and inequities related to mental health in these communities.[24]

Provider pluralism is common in many cities in LMICs and some governments have encouraged the adoption of pluralistic models for healthcare delivery in slums, establishing public–private partnerships between non-governmental organisations (NGOs), government and private providers. The Urban Primary Healthcare Programme project, implemented by the government of Bangladesh, has positively influenced health outcomes for the urban poor in the country.[25] NGOs have frequently led the way in providing healthcare services for slum-dwellers, leading to some reduction in intraurban disparities, particularly in terms of reproductive health.[26]

While this pluralism of providers may improve physical access of poor urban communities to healthcare, several studies have described the barriers faced by slum-dweller on the pathway to health services. Contentious relationships between city authorities and slums-dwellers,[27] and scant public health services whose the access needs to be negotiated through a 'middlemen'[28] have been reported

in several slums. High out of pocket expenditure of urban residents, limited coverage of health insurance and access to subsidised government schemes, particularly among migrants who frequently lack the required registration documents[29–31] are also common. In addition, important differences in costs of accessing healthcare for chronic conditions were found among and within slums in Bangladesh. Slum-dwellers in Tongi community incurred double the costs of those residents of Sylhet (US$323 vs US$169, 2014 prices). While the wealthiest households (fourth and fifth quintiles) were more likely to incur higher costs compared with the poorest households (first and second quintiles) at the same slum.[31] The prevalence of catastrophic health expenditure (CHE) has has also been reported as high. The degree of CHE varies according to the threshold and methods adopted to calculate this indicator.[32] The economic and social impacts of seeking healthcare for slum-dwellers may be further exacerbated by the COVID-19 outbreak due to the severity of the symptoms and its long-term consequences. Several reports have shown high incidence and mortality due to COVID-19 in slums areas of Chembur, Matunga and Dahisar in Mumbai, India.[33 34]

By understanding the structure of expenditure of urban poor residents on healthcare, we will be able to inform decisions on how best to remodel urban health systems and improve equitable access. The objective of this scoping review is therefore to map the evidence available on the economic impact of seeking healthcare on slum-dwellers and the urban poor compared with other city residents. We will quantify the economic impact in terms of CHE, direct and indirect costs incurred by these communities during the search for healthcare and treatment and identify any inequities with better-off urban residents.

## METHODS

This scoping review will be developed as part of the accountability and responsiveness of slum-dwellers (ARISE) research consortium, funded by the Global Challenges Research Fund. The consortium aims to enhance accountability and improve the equitable health and well-being of marginalised populations living in slums in LMICs. The review is part of the ARISE work package Metrics, Epidemiology and Economics, which aims to collect metrics to reflect lived realities, inequities and priorities for transformative social changes in slums by using epidemiological and health economics tools.[35]

This review protocol follows the framework suggested by Arksey and O'Malley[36] and Levac et al[37] and it was registered at the Research Registry (https://www.researchregistry.com/), ID: reviewregistry947. The review process and searches began in June 2020 and we aim to publish our findings by June 2022.

## Box 1 Glossary of operational definitions

Low-income and middle-income countries: Gross national income (GNI) per capita of US$1035 or less for low income economies; GNI per capita between US$1036 and US$4045 for lower-middle-income economies; and GNI per capita between US$4046 and US$12 535 for upper middle-income economies.[3]

Urban informal settlements, slums: households living in urban areas who lack at least one of the following: security of housing tenure, easy access to water or sanitation, sufficient living space, durable housing.[46]

Economic impact: in this review, the economic impact is defined as the financial impact in terms of prevalence of catastrophic health expenditure, direct, indirect and total costs.

Catastrophic health expenditures (CHE): proportion of the household annual income committed with healthcare that results in financial hardship for the family. WHO indicates a threshold of 40% of a household's non-subsistence income.[47] However, other thresholds have been applied in the calculation of CHE.

Direct costs: all costs due to resource use that are completely attributable to the use of a healthcare intervention or illness.[48]

Direct medical costs: out of pocket expenses incurred with medication, tests and healthcare facilities fees.[48]

Direct non-medical costs: out of pocket expenses incurred with transportation, food, accommodation and caregivers.[48]

Indirect costs: income and time lost during the search for healthcare, while waiting for appointments and during hospitalisation.[48]

### Identifying the research question

We formulated our research question based on the ARISE principle to generate evidence to promote (ARISE, 2019). The PCC (population, concept and context) framework was used to formulate the research question.[38] We will explore the concept of the economic impact of seeking healthcare on slum communities compared with other city-residents in the context of urban areas in LMICs. Box 1 shows the glossary of operational definitions adopted in this review.

### Search strategy and identification of relevant studies

The search strategy aimed to identify studies examining the costs of healthcare for slum-dwellers or city residents in LMICs. A search strategy was developed by an information specialist using Ovid MEDLINE and based on the review eligibility criteria indicated in table 1.

Search terms were gathered by applying key concepts: slum-dwellers, slums, informal settlements and urban areas (population), healthcare costs (concept) and LMICs (context). Both text word and subject heading searches for each concept were included in the strategy. Retrieval was limited to publications within the last 10 years (2010–2020). No language restrictions were applied. The final search strategy for MEDLINE was agreed by NTdS-F and HE and then translated as appropriate for the other databases and resources. The complete search strategy is presented in online supplemental material 1.

Databases containing literature from the fields of health, economics and social science was searched in MEDLINE (Ovid), Embase (Ovid), EconLit (Ovid), Science Citation Index (Web of Science), Social Science Citation Index (Web of Science) and Global Index Medicus. Also, the following resources containing grey literature such as theses and dissertations, and reports from leading organisations on urban health research were searched: Proquest Dissertations and Theses (A&I), Econpapers, OpenGrey, the World Bank, the Organisation for Economic Co-operation and Development and The UN-Habitat websites. EndNote was used for reference management and duplicate removal.

### Selection of papers and data extraction

Two reviewers will independently screen the studies by title and abstract generating two lists of retrieved studies to be compared following the pathway indicated on figure 1. The number of records identified, duplicates removed, titles and abstracts screened, studies retrieved and included in the review will be recorded in a Preferred Reporting Items for Systematic Reviews and Meta-Analyses extension for Scoping Reviews (PRISMA) flow chart.[39] Rayyan software (http://rayyan.qcri.org)[40] will be used to manage the screening of all identified studies.

The data extraction will also be conducted by two reviewers, who will extract the data using a predesigned extraction form in the format of Excel spreadsheet.

| Table 1 | Inclusion and exclusion criteria of the scoping review | |
| --- | --- | --- |
| **Appraisal** | **Inclusion criteria** | **Exclusion criteria** |
| Country | Low-income and middle-income countries. | High-income countries. |
| Article type | Peer-reviewed published articles, theses and reports from the World Bank, Organisation for Economic Co-operation and Development and UN-Habitat. | Case reports, protocols, news article, editorial, conference abstracts, comments. |
| Study design | Economic evaluations, cost and cost-effectiveness studies. | Willingness to pay, health finance and economic evaluations addressing the provider perspective, systematic and scoping reviews. |
| Focus of study | Access to healthcare: NGOs, private and public sector, outpatient and inpatient care. Studies that disaggregate by slum/non-slum or wealth quintile/poor/non-poor; studies that are slum specific. | Studies that do not disaggregate between rural and urban or urban wealth categories/slum and city level; studies focused on rural areas. |

NGOs, non-governmental organisations.

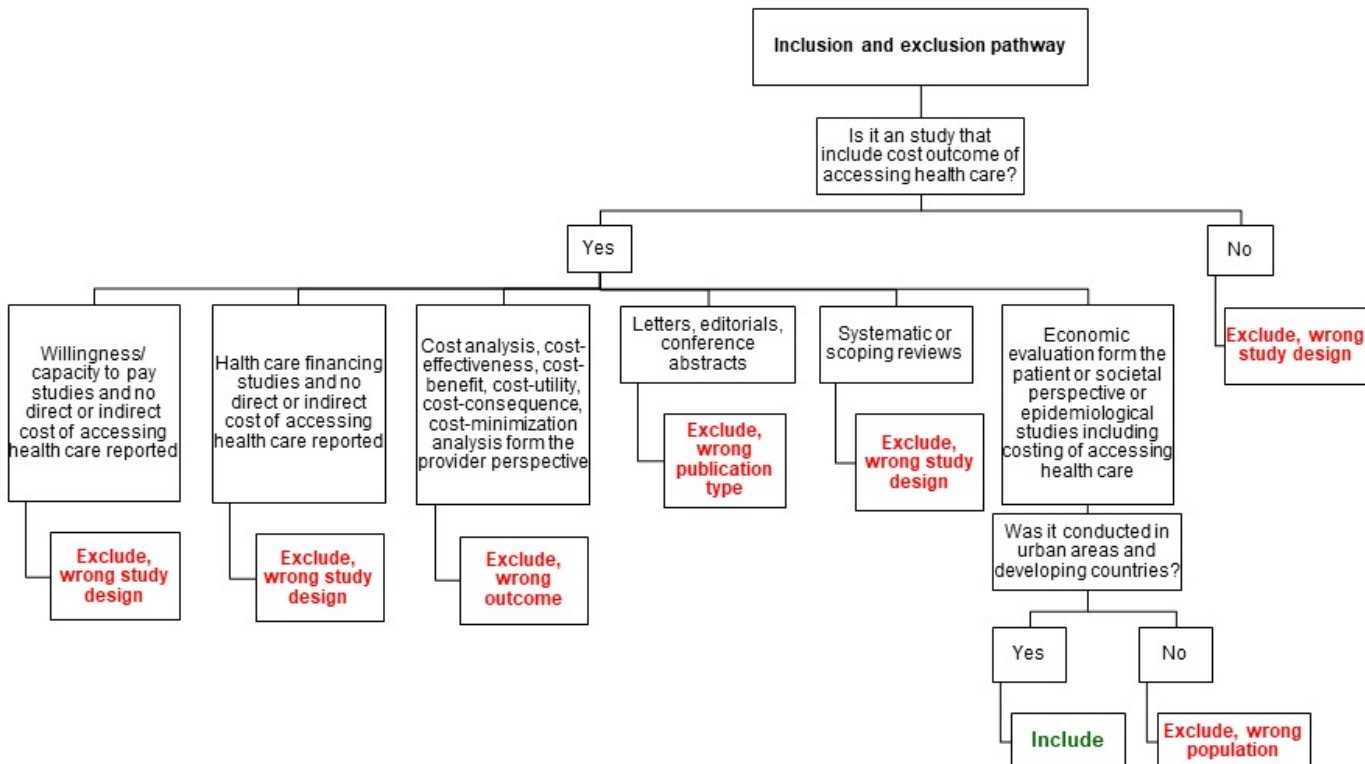

**Figure 1** Inclusion and exclusion pathway of the scoping review.

Discordances in the inclusion of studies and data extraction will be discussed with a third researcher and agreement reached by consensus. We will extract the following information: location (country/city), characteristics of the health sector, socioeconomic profile of the study population, study design, sampling method and sample size, measure of spending/payment, type of costs included, cost analysis and method applied to calculate catastrophic costs, types of healthcare services used, types of health conditions, quality of healthcare, study outcomes (costs and CHE). Data will be classified into the following cost components: direct medical costs (consultations, tests, medicines and hospitalisation, etc), direct non-medical cost (transport and food during healthcare visits) and indirect costs (income and time lost). If data allow, costs will be stratified by socioeconomic status, sex, age, ethnicity, health condition, kind of health service sought and obtained, and regions (slum, non-slum, other). We will extract data on both relative and absolute difference in costs between slum/non-slum and where possible stratified by socioeconomic status. This data extraction strategy will be piloted and adjusted, if necessary.

### Quality assessment

Although quality assessment is not a mandatory step for scoping reviews, we will examine the methodological quality of the studies by using checklists recommended for costing and cost-effectiveness studies.

We will assess the quality of studies based on the Consolidated Health Economic Evaluation Reporting Standards.[41] We will also use the Tool to Estimate Patient's

Costs[42] to evaluate methods applied to estimate costs and procedures adopted for interviewing patients. The quality assessment will be focused on the methods reported in the studies through the assessment of the criteria indicated in table 2.

### Collating, summarising, and reporting the data

Costs reported before 2020 will be updated using USD inflation rates.[43] Costs reported in local currency will then be converted to US Dollars using exchange rates as reported in the OANDA website (OANDA, 2018). To allow comparison of results reported by different countries, costs will be presented in International Dollars ($) applying purchase power parity,[44] 2020 prices. The conversion to international dollars will apply World Bank indices.[45] Results will be presented as the average cost of healthcare, that is, cost of illness and cost of seeking care, comparing slum-dwellers vs other city residents. We will first estimate average total costs and average direct medical, non-medical and indirect costs. When available, we will also estimate disaggregated data on cost by health sector (public/private/NGO; Formal vs Informal), by wealth quintile or household assets, and by disease profile (NCDs, communicable diseases, accidents and violence). The range of reported means across studies, unweighted average of means (with SD), and the median of means (with IQR) will be presented for each costing category. The prevalence of CHE will be reported as range, unweighted average, and median. For studies reporting data from several countries, each country will be analysed as a separate observation. We will run a sensitivity analysis

**Table 2** Items included in the quality assessment of the manuscripts

| | |
|---|---|
| Consolidated health economic evaluation reporting standards | Cost estimation methods. |
| | Target population. |
| | Setting and location. |
| | Study perspective. |
| | Sources used for resource quantities and unit costs. |
| | Period for resources estimation; quantities and unit costs |
| | Methods for adjusting unit costs to the reporting year and performing currency conversion |
| | Values for main categories of estimated costs |
| | Choice of discount rate(s) used for costs and outcomes and why (studies with time horizon above 1 year are expected to discount rates to adjust costs estimates) |
| | Time horizon over which costs and consequences are being evaluated and why |
| Tool to estimate patient's costs | Patient interview procedures |
| | Methods used for valuing indirect costs |

including only studies classified as good quality to evaluate uncertainties in our estimates.

We will report our review following the guideline PRISMA extension for Scoping Reviews (PRISMA-ScR) Checklist (Tricco *et al*). Results will be synthesised and presented using a narrative and tables.

### Ethics
Ethical approval is not required for scoping reviews.

### Patient and public involvement
We formulated our research question based on the ARISE principle to generate evidence to promote ARISE. Results generated by this review will potentially guide the decision-making process to improve access to healthcare and alleviate the economic impact of diseases on slum communities.

Our review team includes participants from all ARISE partners: Liverpool School of Tropical Medicine, University of York and University of Glasgow in UK; Brac University in Bangladesh; LVCT Health and African Population and Health Research Center in Kenya; COMAHS: University of Sierra Leone in Sierra Leone; The George Institute for Global Health and The Society for Promotion of Area Resource Centres in India. The review has a strong capacity strengthening element and all reviewers are

completing the Cochrane interactive learning modules (https://training.cochrane.org/interactivelearning). The team is also meeting regularly to develop skills and ensure quality and consistency throughout the review.

Patients are not directly involved in the review process as this is based on secondary data.

### Dissemination
We will disseminate the results of this review alongside the events organised by the ARISE hub, such as project meetings, seminars, webinars and capacity building training programme. Additionally, we will produce policy briefs including key results of the review, knowledge gaps and priority actions for representatives of the slum communities, policy-makers and other stakeholders in the countries where the ARISE project is working (India, Bangladesh, Sierra Leone, Kenya).

This scoping review addresses a topic of interest to the Sustainable Development Goals (SDG), particularly SDG 3, Good Health and Well-being, and 10, Reduce Inequalities. Thus, our review will be disseminated at international conferences and will potentially contribute to formulating public policies to improve access to healthcare by slum-dwellers.

The final manuscript will be submitted to an open-access international journal to reach a larger audience.

**Author affiliations**
[1]Department of Health Sciences, University of York, York, UK
[2]MRC/CSO Social and Public Health Sciences Unit, University of Glasgow, Glasgow, UK
[3]BRAC University James P Grant School of Public Health, Dhaka, Bangladesh
[4]Department of Clinical Sciences, Liverpool School of Tropical Medicine, Liverpool, UK
[5]LVCT, Nairobi, Kenya
[6]COMAHS, Freetown, Sierra Leone
[7]The George Institute for Global Health India, New Delhi, India
[8]African Population and Health Research Center, Nairobi, Kenya
[9]Clinical Sciences and International Public Health, Liverpool School of Tropical Medicine, Liverpool, UK
[10]Centre for Health Economics, University of York, York, UK
[11]The Society for Promotion of Area Resource Center, Mumbai, India
[12]School of Public Health, BRAC University James P Grant School of Public Health, Dhaka, Bangladesh

**Acknowledgements** We thank Melissa Harden, the information specialist form the Centre for Reviews and Dissemination at the University of York, for developing the search strategy.

**Contributors** The review team is distributed in three groups and participants were allocated to these groups according to their skills, experience in conducting systematic reviews and data analysis, and time availability. NTdS-F, HE, JL, EK, PP-H and ZQ are part of the core group and are responsible for developing the main activities of the scoping review, such as screen and select the manuscripts, extract the data and data analysis and writing the manuscript and reports. The second group comprises new reviewers. They will participate in the whole review process under the guidance and mentorship of the participants from the core group. AA, MIHM, FM, RK, SS, JS, VS, SG and IC are part of the new reviewer group. The third group comprises the advisory group and includes senior researchers with expertise in health economics, urban health and slums. They will guide the core team on the data analysis process, definitions of keyword, such as slums and economic impact, also acting as tie-breakers on the occasion of disagreements in the inclusion and data extraction stages of the review. BM, VR, RT and SM are part of this group. This

protocol was written by NTdS-F and HE with contributions and approval from all coauthors.

**Funding** This review is supported by ARISE UK Research and Innovation's Global Challenges Research Fund (ES/S00811X/1, ARISE's Metrics, Epidemiology and Economics work package group/University of York). EK also acknowledges support from the Medical Research Council and Scottish Government Chief Scientist Office funding (MC_UU_00022/2, SPHSU17).

**Disclaimer** The funders had or will have no role in the development of this protocol, the collection and analyses, or interpretation of results, or in the writing or publication of the review's results.

**Competing interests** None declared.

**Patient consent for publication** Not required.

**Provenance and peer review** Not commissioned; externally peer reviewed.

**ORCID iDs**
Noemia Teixeira de Siqueira-Filha http://orcid.org/0000-0003-0730-8561
Jinshuo Li http://orcid.org/0000-0003-1496-7450
Eliud Kibuchi http://orcid.org/0000-0002-5091-6450
Md Imran Hossain Mithu http://orcid.org/0000-0001-8057-3092
Sumit Mazumdar http://orcid.org/0000-0002-9278-4396

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
