## [Reviewer comments · BMJ Open]

ARTICLE DETAILS

TITLE (PROVISIONAL)	The economics of health-care access in low and middle-income countries: A protocol for a scoping review of the economic impacts of seeking health care on slum-dwellers compared with other city residents.
AUTHORS	Teixeira de Siqueira-Filha, Noemia; Li, Jinshuo; Kibuchi, Eliud; Quayyum, Zahidul; Phillips-Howard, Penelope; Awal, Abdul; Mithu, Md. Imran Hossain; Manzoor, Farzana; Karuga, Robinson; Saidu, Samuel; Smith, John; Sai, Varun; Garimella, Sureka; Chumo, Ivy; Mberu, Blessing; Tolhurst, Rachel; Mazumdar, Sumit; Rao, Vinodkumar; Farnaz, Nadia; Alam, Wafa; Elsey, Helen

VERSION 1 – REVIEW

REVIEWER	Boone, Jan Tilburg University, Economics
REVIEW RETURNED	30-Dec-2020

GENERAL COMMENTS	Referee report on "The economics of healthcare access in low and middle-income countries: A protocol for a scoping review of the economic impacts of seeking health care on slum-dwellers compared with other city residents." Summary This paper proposes a literature study comparing access to healthcare for sum-dwellers compared to other city residents in low and middle-income countries. This is a relevant and interesting topic. I would be interested in such a literature study, but I do have some questions/comments. Comments  the Editors' instructions mention that the dates of the study should be included in the manuscript. I was not able to find these. section 1 starts with a motivation of the study which refers to urbanization as a driver of growth as well as inadequate access to healthcare (and other problems). This is slightly confusing as urbanization invites a comparison between rural and slum access to healthcare (see also line 15 on page 4) while the comparison actually is between slum-dwellers and other city residents. proposal focuses on catastrophic health expenditure (CHE); box 1 defines economic impact as "the financial impact in terms of prevalence of catastrophic health expenditure, direct, indirect and total costs". My main question refers to the indirect costs (see also page 7, line 43). This seems to refer to indirect costs (lost income etc.) related to treatment. If this is correct, there would be a sizable under-estimation of the true costs for slum dwellers. To illustrate what I have in mind, consider the extreme of a slum where there is no medical care at all available. Clearly, CHE would be close to zero: if you cannot spend money on treatment, there is no CHE. But the costs of illness (in terms of
--

	utility lost –perhaps measured in qaly’s– and income lost due to reduced productivity) would be huge. Will such indirect costs of illness (not treatment) be accounted for in the literature study? Page 8, line 36 "Results will be presented as the average cost of seeking health care 1 comparing slum dwellers vs other city residents." suggests not. This could be problematic.  • a related point is the comparison between slum dwellers and other city residents. One should be careful in interpreting the results of such a comparison. A policy that increases costs for other city residents (but not for slum dwellers) does not improve the situation of slum dwellers (they are relatively better off, but not absolutely better off). Perhaps some surveyed papers report both relative and absolute comparisons for slum-dwellers. If this is the case, it would be useful to take note of this difference. • page 7, line 38, the information extracted from papers includes study design. Does this also include the identification strategy of estimating costs? Some studies can be more convincing than others on this front (next to sampling issues etc.). Minor comments  • page 4, line 9: "absolute number of 881,080 people is close to the entire population of the two most populous countries in the world, India (1.4 billion inhabitants) and China (1.3 billion inhabitants)" How is 881,000 close to 1 billion? • page 4, line 39, typo: "stablishing" • page 8, line 36, typo: "conversation" of currencies
--	--

VERSION 1 – AUTHOR RESPONSE

Comments

1. the Editors’ instructions mention that the dates of the study should be included in the manuscript. I was not able to find these.

We have included this information at the end of paragraph two of the methods section: The review process and searches began in June 2020 and we aim to publish our findings by June 2022.

2. section 1 starts with a motivation of the study which refers to urbanization as a driver of growth as well as inadequate access to healthcare (and other problems). This is slightly confusing as urbanization invites a comparison between rural and slum access to healthcare (see also line 15 on page 4) while the comparison actually is between slum-dwellers and other city residents.

Thanks for your comment. We are interested in understanding how urbanization can increase inequalities in health care access within the urban context. Our hypothesis is that more vulnerable populations, both slum and non-slums residents, can incur the more severe economic burden of accessing health care when compared with those who are better off and living in the same urban context. Thus, we will compare slum vs non-slum urban residents in our review in order to identify any inequities within cities. We have added to the introduction section in order to clarify this point.

3. proposal focuses on catastrophic health expenditure (CHE); box 1 defines economic impact as "the financial impact in terms of prevalence of catastrophic health expenditure, direct, indirect and total costs". My main question refers to the indirect costs (see also page 7, line 43). This seems to refer to indirect costs (lost income etc.) related to treatment. If this is correct, there would be a sizable under-

estimation of the true costs for slum dwellers. To illustrate what I have in mind, consider the extreme of a slum where there is no medical care at all available. Clearly, CHE would be close to zero: if you cannot spend money on treatment, there is no CHE. But the costs of illness (in terms of utility lost – perhaps measured in qaly's– and income lost due to reduced productivity) would be huge. Will such indirect costs of illness (not treatment) be accounted for in the literature study? Page 8, line 36 "Results will be presented as the average cost of seeking health care 1 comparing slum dwellers vs other city residents." suggests not. This could be problematic.

Thanks for pointing out this limitation. Our review also covers studies addressing cost of illness (see search strategy in the supplementary material, page 4, line 22) and access to informal health care, such as traditional healing and self-medication, which is quite common among the poorest populations. We will not include utility lost in terms of QALY as we want to focus on the economic burden in monetary values, that is costs (direct and indirect) and catastrophic costs. Also, the inclusion of QALY as an outcome would give us a broad range of publications which would be difficult to manage given the time available to complete this review. We have clarified the statement on page 8, *"Results will be presented as the average cost of health care, that is cost of illness and cost of seeking care, comparing slum dwellers vs other city residents."*

4. a related point is the comparison between slum dwellers and other city residents. One should be careful in interpreting the results of such a comparison. A policy that increases costs for other city residents (but not for slum dwellers) does not improve the situation of slum dwellers (they are relatively better off, but not absolutely better off). Perhaps some surveyed papers report both relative and absolute comparisons for slum-dwellers. If this is the case, it would be useful to take note of this difference.

Thanks for this important comment. We will extract data on both relative and absolute difference in costs between slum/non-slum from included papers and ensure this is reported in the presentation of findings. This has been clarified in the methods section under data extraction. We aim to use the results of our review to advocate for the improvement of health care access for slum-dwellers or low-income urban residents by reducing inequalities through the provision of health insurance, free access to health care, and reductions of barriers on the pathway of seeking care. We will also measure the economic burden by wealthy quintile to take into account these inequalities for both slum-dwellers and other city residents.

5. page 7, line 38, the information extracted from papers includes study design. Does this also include the identification strategy of estimating costs? Some studies can be more convincing than others on this front (next to sampling issues etc.).

That is correct, the data extraction form includes information on measure of spending/payment , type of costs included, cost analysis, and method applied to calculate catastrophic costs. We will also consider these issues in the data analysis by running a sensitivity analysis. We have included this information in page 7 and page 8 (Collating, summarizing, and reporting the data).

Minor comments

6. page 4, line 9: "absolute number of 881,080 people is close to the entire population of the two most populous countries in the world, India (1.4 billion inhabitants) and China (1.3 billion inhabitants)" How is 881,000 close to 1 billion?

Yes, this is an error, the correct number is 881,080,000 people. We have corrected it on page 5.

7. page 4, line 39, typo: "stablishing"

8. page 8, line 36, typo: "conversation" of currencies

Thanks for pointing out these typos, we have corrected them in the text.